# Automated Analysis vs. Expert Reading in Nuclear Cardiology: Correlations with the Angiographic Score

**DOI:** 10.3390/medicina58101432

**Published:** 2022-10-11

**Authors:** George Angelidis, Varvara Valotassiou, Ioannis Tsougos, Chara Tzavara, Dimitrios Psimadas, Evdoxia Theodorou, Anastasia Ziaka, Stavroula Giannakou, Charalampos Ziangas, John Skoularigis, Filippos Triposkiadis, Panagiotis Georgoulias

**Affiliations:** 1Nuclear Medicine Laboratory, University of Thessaly, University Hospital of Larissa, Mezourlo, 41110 Larissa, Greece; 2Medical Physics Laboratory, University of Thessaly, University Hospital of Larissa, Mezourlo, 41110 Larissa, Greece; 3Department of Cardiology, University of Thessaly, University Hospital of Larissa, Mezourlo, 41110 Larissa, Greece

**Keywords:** automated analysis, myocardial perfusion imaging, summed difference score, summed rest score, summed stress score, coronary angiography

## Abstract

*Background and Objectives*: Myocardial perfusion imaging (MPI) has an important role in the non-invasive investigation of coronary artery disease. The interpretation of MPI studies is mainly based on the visual evaluation of the reconstructed images, while automated quantitation methods may add useful data for each patient. However, little evidence is currently available regarding the actual incremental clinical diagnostic performance of automated MPI analysis. In the present study, we aimed to assess the correlation between automated measurements of Summed Stress Score (SSS), Summed Rest Score (SRS) and Summed Difference Score (SDS), with the corresponding expert reading values, using coronary angiography as the gold standard. *Materials and Methods*: The study was conducted at the Nuclear Medicine Laboratory of the University Hospital of Larissa, Larissa, Greece, οver an one-year period (January 2019–January 2020). 306 patients, with known or suspected coronary artery disease, were enrolled in the study. Each participant underwent a coronary angiography, prior to or after the scintigraphic study (within a three-month period). Either symptom-limited treadmill test, or pharmacologic testing using adenosine or regadenoson, was performed in all participants, and the scintigraphic studies were carried out using technetium 99m (99mTc) tetrofosmin (one-day stress/rest protocol). Coronary angiographies were scored according to a 4-point scoring system (angiographic score; O: normal study, 1: one-vessel disease, 2: two-vessel disease, 3: three-vessel disease). Moreover, automated measurements of SSS, SRS and SDS were derived by three widely available software packages (Emory Cardiac Toolbox, Myovation, Quantitative Perfusion SPECT). *Results*: Interclass Correlation Coefficients of SSS, SRS and SDS between expert reading and software packages were moderate to excellent. Visually defined SSS, SRS and SDS were significantly correlated with the corresponding results of all software packages. However, visually defined SSS, SRS and SDS were more strongly correlated with the angiographic score, indicating a better performance of expert reading when compared to automated analysis. *Conclusions*: Based on our results, visual evaluation continues to have a crucial role for the interpretation of MPI images. Software packages can provide automated measurements of several parameters, particularly contributing to the investigation of cases with ambiguous scintigraphic findings.

## 1. Introduction

Single photon emission computed tomography (SPECT) myocardial perfusion imaging (MPI) represents one of the most widely used imaging modalities for the non-invasive investigation of coronary artery disease (CAD) [1]. In general, myocardial SPECT results are based on the visual evaluation of the reconstructed images. The reader should also take into account several additional factors, such as the pre-test likelihood of disease, image quality and the potential presence of artefacts.

An interesting factor of SPECT MPI is the development of standardised methods for automated quantitation. Currently, automated analysis of three-dimensional myocardial SPECT data is a constant component of practice in nuclear cardiology. To assist clinical decision making, commercially available software packages can provide myocardial perfusion maps and estimate global and segmental measures of stress/rest perfusion. Regional perfusion scores (17-segment model) can be derived using the average defect severity in a given segment [2]. Software packages assign severity scores to segments, according to a 5-point scale [3]. Segmental scores can be summed either per region, or for the whole myocardium [summed stress score (SSS), summed rest score (SRS), summed difference score (SDS)].

Software packages have been developed using normal databases [4,5,6]. However, evidence regarding the association between automated quantitation and expert reading (ER) is scarce [7,8,9]. Therefore, the actual incremental clinical diagnostic performance of automated SPECT MPI analysis remains rather unclear. In the present study, we aimed to investigate the correlation between SSS, SRS and SDS values, derived by three widely available software packages [Emory Cardiac Toolbox (ECTb), Myovation (MYO), Quantitative Perfusion SPECT (QPS)], with the reader scoring of these parameters. Subsequently, we assessed the associations between the SSS, SRS, and SDS values, as recorded by automated analyses and reader scoring, with angiographic score, using coronary angiography as the gold standard.

## 2. Materials and Methods

### 2.1. Study Population

The present study was conducted at the Nuclear Medicine Laboratory of the University Hospital of Larissa, Larissa, Greece. Over the 1-year study period (January 2019–January 2020), 1063 patients with known or suspected (medium or high risk stratification) CAD were referred to our laboratory for stress/rest myocardial perfusion SPECT. Out of the patients who underwent coronary angiography prior to or after SPECT MPI (within a 3-month period), 306 consecutive patients were enrolled in the study, as they did not meet any of the exclusion criteria. Before testing, a brief structured interview took place, and data regarding clinical features, medications, previous cardiac events, CAD risk factors, and cardiac or noncardiac comorbidities were collected. Hypertension was defined as a systolic blood pressure of 140 mmHg or greater at rest and/or a diastolic blood pressure of 90 mmHg or greater at rest, or treatment with antihypertensive agents. Furthermore, patients with diabetes mellitus or lipid disorders were defined according to the interview data, including the use of the corresponding medications. Obesity was considered as a condition with body mass index (BMI) of 30.0 or greater (BMI calculated as weight in kilograms divided by height in metres squared).

Due to their potential effects on performance during stress testing, MPI, and the associated parameters, cardio-active medications (i.e., b-blockers, calcium channel antagonists, and nitrates) were temporarily withdrawn for approximately five half-lives [3,10]. Patients without proper withdrawal of cardio-active medications were not included into the study. Moreover, we excluded patients with severe congenital or valvular heart disorder, as well as those with non-ischemic cardiomyopathy. Other exclusion criteria were previous cardiac invasive procedure [percutaneous coronary intervention (PCI) or coronary artery bypass grafting (CABG)], or a history or other evidence of myocardial infarction. Patients with a previous myocardial infarction comprise an heterogeneous group whose myocardial perfusion study is influenced not only by myocardial ischemia, but also by necrosis linked to both episode severity and applied therapy. Finally, patients with qualitatively suboptimal scintigrams, due to artefacts, were excluded from the study. 

There was a contraindication or inability to achieve a satisfactory exercise level in 183 patients. In these patients, either adenosine or regadenoson pharmacologic testing, combined with low-level exercise, was performed. Moreover, pharmacologic testing without any type of exercise was performed in nine patients with left bundle branch block (LBBB) or an implantable pacemaker. 

All participants gave informed consent for their complete participation in the study, according to the Hospital Ethics Committee guidelines and in compliance with the ethical guidelines of the Declaration of Helsinki. Moreover, all participants were given written information concerning the appropriate radiation protection measures.

### 2.2. Stress Testing

Patients underwent symptom-limited treadmill tests according to the Bruce Protocol, after cardio-active medication withdrawal, 6 h- to 12 h-fasting and avoiding smoking or engaging in intense physical activity for at least 3 h before the examination, as previously described [10]. Data on symptoms related to the performance of exercise testing, and estimated workload in metabolic equivalents (METs, using standard tables) were recorded. Adenosine or regadenoson pharmacologic testing, combined with low-level exercise or not, was performed based on the European Association of Nuclear Medicine (EANM) procedural guidelines [10,11].

### 2.3. Coronary Angiography and Angiographic Score

Cardiac catheterization studies were requested by cardiologists based on patients’ data. All coronary angiographies were blindly interpreted by one experienced observer. Each stenosis of the vessel lumen greater than 50% was considered hemodynamically significant, while the presence of a stenosis in the left mainstem was considered equivalent to a two-vessel disease. Therefore, each angiographic study was scored according to a 4-point scoring system (angiographic score; O: normal study, 1: one-vessel disease, 2: two-vessel disease, 3: three-vessel disease).

### 2.4. SPECT Myocardial Perfusion Imaging and Semi-Quantification

Acquisition and processing protocols were in accordance with the EANM/European Society of Cardiology (EANM/ESC) procedural guidelines [10]. The range of injected activities was 250–400 MBq for stress acquisitions and 625–1000 MBq for rest acquisitions. Studies were performed using technetium 99m (99mTc) tetrofosmin (Myoview, GE Healthcare, Chicago, IL, USA). All acquisitions were carried out in the supine position, without attenuation-scatter correction, using a dual-headed SPECT camera. Polar and three-dimensional mapping were performed (GE Xeleris Software, Chicago, IL, USA), and filtered back projection with the Butterworth Filter was used for tomographic reconstruction.

Polar maps, three-dimensional images, and the reconstructed images of both stress and rest studies were blindly evaluated by two independent experienced observers. Left ventricular (LV) myocardium was divided into 17 segments and radiotracer uptake was scored in each of these segments according to a 5-point scoring system (0: normal uptake; 1: mildly decreased uptake; 2: moderately decreased uptake; 3: severely decreased uptake and 4: no uptake) [2]. If counts were reduced in a region and this was attributed to attenuation artefact, the score was 0 [12]. The view of a third observer was requested in eight studies in which discordance between the two observers was detected, and the disagreement was resolved by consensus [13]. SSS and SRS were calculated by adding the scores of each segment in stress and rest studies, and SDS was obtained by subtracting SRS from SSS [2]. The level of inter-rater agreement for SSS, SRS and SDS was significant, with intraclass correlation coefficients (ICCs) ranging from 0.89 to 0.93, (*p* < 0.001).

Moreover, for each participant, SSS, SRS and SDS values were recorded, as derived by ECTb and QPS software packages. MYO does not provide standardised segmental perfusion scores. Therefore, we converted the average segmental count values (relative to maximum pixel values in the relevant polar plot) to categorical scores according to >70%, 50–69%, 30–49%, 10–29% and <10% thresholds, as previously described [14]. Finally, automated myocardial perfusion measurements were based on the results of the commercially available packages, without any institutional adjustments.

## 3. Statistical Analysis

Quantitative variables were expressed as mean (standard deviation) or as median (interquantile range). Qualitative variables were expressed as absolute and relative frequencies. Spearman correlation coefficients were used to explore the association of two continuous variables. Correlation coefficients between 0.1–0.29 were considered “poor”, between 0.30–0.49 “fair”, between 0.50–0.69 “moderate” and between 0.70–1.0 “strong”. ICC was used to assess inter-rater agreement, as well as the agreement between ER and the software packages, concerning SSS, SRS and SDS parameters. Following the recommendations given by Koo and Li, ICC below 0.5 indicates poor agreement, between 0.5 and 0.75 indicates moderate agreement, between 0.75 and 0.90 indicates good agreement, and above 0.9 indicates excellent agreement [15]. Agreement between the expert and the software packages was further assessed by Bland–Altman 95% confidence intervals (CI) for limits of agreement (LOA). The 95% CI for LOA indicates that 95% of the differences fall between these two limits. All reported *p* values are two-tailed. Statistical significance was set at *p* < 0.05 and analyses were conducted using SPSS statistical software (version 22.0).

## 4. Results

The sample consisted of 306 patients (62.7% male) with mean age 63.8 years (SD = 9.9 years). Demographic and clinical characteristics of the patients are presented in Table 1. The majority of the patients reported having CAD-related symptoms (65.4%). Obesity was recorded in 45.2% of the sample. Smokers were 41.2% of the patients. Arterial hypertension and diabetes mellitus were recorded in 77.8% and 36.6% of the patients, respectively. Median angiographic score was 1 (IQR: 0–2).

Mean SSS and mean SRS according to ER were significantly lower than the corresponding SSS and SRS values derived from ECTb, MYO and QPS (Table 2). Mean SDS based on ER was significantly lower than SDS derived from ECTb and MYO, but significantly higher in comparison to QPS results. ICCs of SSS, SRS and SDS between ER and the software packages are presented in Table 3. All ICCs were moderate to excellent, as well as significant (SSS ICCs ranged from 0.82 to 0.91, SRS ICCs ranged from 0.67 to 0.79 and SDS ICCS ranged from 0.69 to 0.82).

Bland-Altman plots for SSS, SRS and SDS are presented in Figure 1. For SSS, limits of agreement between ER and ECTb ranged from −5.07 to 11.82 (Figure 1(A1)), between ER and MYO from −4.41 to 11.24 (Figure 1(A2)) and between ER and QPS from −5.02 to 4.90 (Figure 1(A3)). For SRS, limits of agreement between ER and ECTb ranged from −2.70 to 7.81 (Figure 1(B1)), between ER and MYO from −2.54 to 9.31 (Figure 1(B2)) and between ER and QPS from −2.83 to 4.35 (Figure 1(B3)). For SDS, limits of agreement between ER and ECTb ranged from −6.02 to 7.64 (Figure 1(C1)), between ER and MYO from −6.46 to 6.54 (Figure 1(C2)) and between ER and QPS from −5.55 to 3.91 (Figure 1(C3)). Limits of agreement between ER and software packages were widely indicating that the average discrepancy between expert scoring and software analyses was large enough.

SSS values based on ER were significantly correlated with the corresponding results of all software packages (Table 4) and all coefficients were “strong”. Similarly, SRS and SDS values according to ER were significantly associated with SRS and SDS results from all software packages and correlations for SRS were almost all moderate.

Table 5 presents Spearman’s correlation coefficients of angiographic score with SSS, SRS and SDS based on ER, as well as with the corresponding results derived from software packages. SSS, SRS and SDS values according to either software packages or expert reading were significantly associated with angiographic score. By examining the coefficients, however, expert estimation of SSS, SRS and SDS was more strongly correlated with the angiographic score compared to all three software packages. Specifically, correlation coefficients of the angiographic score with the software packages were fair and less than 0.5, while correlation coefficients of the angiographic score with ER were more than 0.5 and moderate.

## 5. Discussion

SPECT MPI is a widely used non-invasive imaging modality for the investigation of patients with known or suspected CAD [16]. In general, MPI results are based on the visual evaluation of the reconstructed images. In an effort to restrict the influence of reader’s experience over image interpretation, a number of related algorithms were developed aiming to provide automated measurements of myocardial perfusion. Nowadays, several software packages, such as ECTb, MYO and QPS, are available, and automated analysis of myocardial perfusion has become part of the routine practice in nuclear cardiology. Interestingly, previous studies have demonstrated that software packages can supplement visual evaluation, as well as their high reproducibility [4,5,6,7,8].

However, automated analysis is currently used only as an adjunct to visual interpretation [7,8,9,17]. The adjunctive role of automated analyses in nuclear cardiology practice is at least partially associated with a main drawback of software packages, which cannot explicitly differentiate between real perfusion abnormalities and artefacts [8]. In addition, although the performance of the software packages has been demonstrated to be similar, certain differences in the magnitudes of the derived values have been reported [18]. In particular, comparing the diagnostic performance of three software packages (ECTb, QPS, and 4DMSPECT), Wolak et al. reported differences in myocardial perfusion quantification, while Knollmann et al. found differences in measurements (derived from QPS and 4DMSPECT), using either manufacturer’s or institutional normal databases [19,20]. Similarly, Johansson et al. demonstrated considerable differences in performance between ECTb, QPS and 4DMSPECT, and Knollmann et al. noted the influence of heart-axis tilt on automated SSS calculations [21,22]. Furthermore, studying a large patient population in our laboratory, we reported a favourable concordance between ECTb, MYO and QPS, whereas differences were demonstrated in pair comparisons [23]. Obviously, for each software package, quantitative measurements are algorithm-specific, and the observed differences in the derived values among software packages could be related to the different principles and assumptions on which each algorithm is based.

Although several studies exist investigating the cross-correlation of the outputs of different quantitative SPECT MPI algorithms, there is only few published data regarding the association between automated quantitation and ER. In the present study, we aimed to investigate the correlation between automated measurements of SSS, SRS and SDS (using three software packages, as commercially available) with the ER of these parameters. Previously, Arsanjani et al. and Duvall et al. had compared the automated measurements of total perfusion deficit (TPD) with the visual quantitation of stress and rest images, while Driessen at al. had investigated MPI SPECT data, both visually and automatically [SSS, SDS, stress total perfusion deficit (S-TPD), and ischemic total perfusion deficit (I-TPD)] [7,8,9].

We demonstrated that the ICCs of SSS, SRS and SDS between ER and the software packages were moderate to excellent, while the average discrepancy between expert scoring and software analyses was large enough (based on the limits of agreement between ER and software packages). Furthermore, although visually defined SSS, SRS and SDS were significantly correlated with the corresponding results of all software packages, expert estimations of SSS, SRS and SDS were more strongly correlated with the angiographic score. On the other hand, Arsanjani et al. reported that automated analysis of attenuation-corrected (AC) and non-corrected (NC) SPECT MPIs was at least equivalent to visual interpretation, when compared to coronary angiography (≥70% luminal stenosis) [8]. Furthermore, Duvall et al. demonstrated that automated analysis and visual quantitation of stress and rest images had similar diagnostic accuracy, in terms of the angiographic detection of ≥70% stenoses, with the most favourable diagnostic accuracy achieved in combined supine and prone stress imaging [7]. Finally, Driessen et al. found that visual quantification slightly outperformed automated analysis in the detection of fractional flow reserve-defined significant CAD, whereas the diagnostic accuracy of automated analysis equalled ER after optimization with an institutional normal database and thresholds [9]. The observed differences between our results and those reported in previous studies could be partially attributed to the differences in study samples, as well as in the definition of the hemodynamically significant stenosis (greater than 50% vs. 70% luminal stenosis). 

The visual evaluation of myocardial SPECT data continues to play a crucial role in the interpretation of MPI examinations. To the best of our knowledge, this is the first study investigating the correlations between automated analysis and ER of SSS, SRS, and SDS, with regard to the angiographic score. Based on the existing research, ER has demonstrated similar performance in the quantification of myocardial perfusion, in comparison to that derived from automated analyses using different methodologies. According to our results, expert estimations of SSS, SRS and SDS were significantly correlated with the corresponding results of all software packages, but visually defined SSS, SRS and SDS were more strongly correlated with the angiographic score, indicating a better performance of ER when compared to automated analysis. However, automated analysis could have an adjunctive role in the interpretation of SPECT MPI examinations, particularly in patients with ambiguous scintigraphic findings. In addition, automated measurements may help beginners, or less experienced physicians, in their practice.

## 6. Conclusions

In the present study, comparing the performance of automated analysis to expert reading, we demonstrated that visually defined SSS, SRS and SDS were more strongly correlated to angiographic score than the software-derived corresponding values. Our results support the significance of ER for the interpretation of myocardial SPECT data. Measurements derived from software packages can be useful mainly in cases of ambiguous findings, supplementary to visual interpretation.

## Figures and Tables

**Figure 1 medicina-58-01432-f001:**
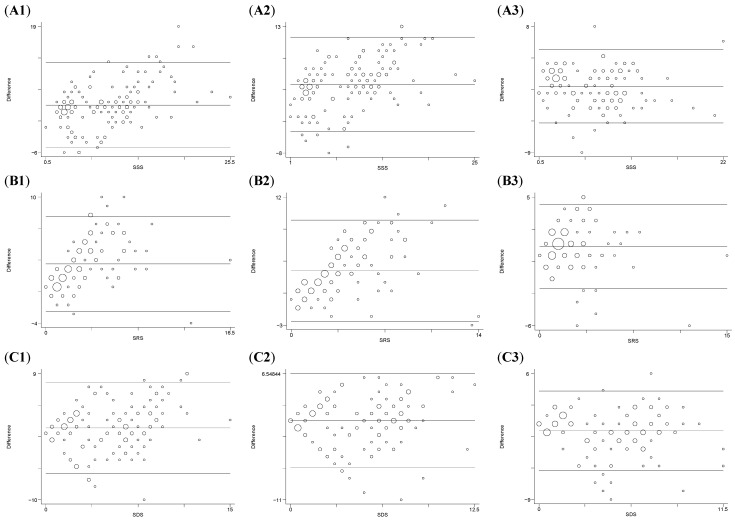
Bland–Altman Plots for (**A**) SSS, (**B**) SRS, and (**C**) SDS. 1, 2, 3 denote ECTb, MYO and QPS software packages, respectively. ECTb: Emory Cardiac Toolbox; MYO: Myovation; QPS: Quantitative Perfusion Single photon emission computed tomography; SDS: summed difference score; SRS: summed rest score; SSS: summed stress score.

**Table 1 medicina-58-01432-t001:** Sample characteristics.

	N (%)
Gender	
Males	192 (62.7)
Females	114 (37.3)
Age, mean (SD)	63.8 (9.9)
Body mass index, mean (SD)	29.8 (4.8)
Body mass index	
Normal	40 (16.1)
Overweight	96 (38.7)
Obese	112 (45.2)
Symptoms	200 (65.4)
Angina	88 (28.8)
Angina-like symptoms	42 (13.7)
Dyspnea	42 (13.7)
Palpitations	58 (19)
Fatigue	44 (14.4)
Smoking	126 (41.2)
Hypertension	238 (77.8)
Diabetes mellitus	112 (36.6)
Lipid disorders	244 (79.7)
Obesity	144 (47.1)
Family history of coronary artery disease	132 (43.1)
Peripheral angiopathy	20 (6.5)
Stroke	26 (8.5)
Chronic obstructive pulmonary disease	32 (10.5)
Previous myocardial infarction	62 (20.4)
Available Echo data	228 (74.5)
LVEF, mean (SD)	0.54 (0.10)
Coronary angiography	306 (100.0)
Percutaneous coronary intervention	0 (0.0)
Coronary artery by-pass grafting	0 (0.0)
Left main artery	0 (0.0)
Left anterior descending	128 (41.8)
Left circumflex	66 (21.6)
Right coronary artery	106 (34.6)
Angiographic score	
Mean (SD)	0.97 (0.99)
Median (IQR)	1 (0–2)
Angiographic score	
0	126 (41.2)
1	88 (28.8)
2	66 (21.6)
3	26 (8.5)
Cardioactive agents	228 (74.5)
Bruce protocol	114 (37)
Pharmacologic testing	192 (63)

LVEF: left ventricular ejection function.

**Table 2 medicina-58-01432-t002:** SSS, SRS, SDS values for ER and ECTb, MYO, and QPS software packages.

	ER	ECTb	MYO	QPS	ECTb vs. ER		MYO vs. ER		QPS vs. ER	
	Mean (SD)	Mean (SD)	Mean (SD)	Mean (SD)	Mean Difference (SD)	*p*	Mean Difference (SD)	*p*	Mean Difference (SD)	*p*
SSS	6.6 (4.4)	10 (6.3)	10 (6.1)	6.8 (4.2)	3.4 (4.2)	<0.001	3.4 (3.9)	<0.001	0.2 (2.5)	0.043
SRS	2 (2.2)	4.6 (3.3)	5.4 (3.6)	2.8 (2.1)	2.6 (2.6)	<0.001	3.4 (3.0)	<0.001	0.8 (1.8)	<0.001
SDS	4.6 (3.1)	5.4 (4)	4.8 (3.6)	3.8 (2.9)	0.8 (3.4)	<0.001	0.2 (3.2)	0.050	−0.8 (2.4)	<0.001

ECTb: Emory Cardiac Toolbox; ER: expert reading; MYO: Myovation; QPS: Quantitative Perfusion Single photon emission computed tomography; SDS: summed difference score; SRS: summed rest score; SSS: summed stress score.

**Table 3 medicina-58-01432-t003:** ICCs for SSS, SRS and SDS between ER and ECTb, MYO, and QPS software packages.

	ECTb vs. ER		MYO vs. ER		QPS vs. ER	
	ICC (95% CI)	*p*	ICC (95% CI)	*p*	ICC (95% CI)	*p*
SSS	0.82 (0.78–0.86)	<0.001	0.85 (0.81–0.88)	<0.001	0.91 (0.89–0.93)	<0.001
SRS	0.73 (0.66–0.78)	<0.001	0.67 (0.59–0.74)	<0.001	0.79 (0.74–0.83)	<0.001
SDS	0.71 (0.63–0.77)	<0.001	0.69 (0.61–0.75)	<0.001	0.82 (0.77–0.85)	<0.001

ECTb: Emory Cardiac Toolbox; ER: expert reading; ICC: intraclass correlation coefficient; MYO: Myovation; QPS: Quantitative Perfusion Single photon emission computed tomography; SDS: summed difference score; SRS: summed rest score; SSS: summed stress score.

**Table 4 medicina-58-01432-t004:** Spearman correlation coefficients of expert estimations of SSS, SRS and SDS and the correspondent values derived from ECTb, MYO, and QPS software packages.

	SSS	SRS	SDS
ECTb	0.72	0.51	0.56
MYO	0.74	0.55	0.54
QPS	0.76	0.46	0.68

Note. All coefficients were significant at *p* < 0.001. ECTb: Emory Cardiac Toolbox; ER: expert reading; MYO: Myovation; QPS: Quantitative Perfusion Single photon emission computed tomography; SDS: summed difference score; SRS: summed rest score; SSS: summed stress score.

**Table 5 medicina-58-01432-t005:** Spearman correlation coefficients of the angiographic score with expert estimations and software-derived SSS, SRS and SDS.

	SSS	SRS	SDS
ECTb	0.44	0.37	0.37
MYO	0.49	0.40	0.40
QPS	0.48	0.31	0.47
ER	0.63	0.51	0.66

Note. All coefficients were significant at *p* < 0.001. ECTb: Emory Cardiac Toolbox; ER: expert reading; MYO: Myovation; QPS: Quantitative Perfusion Single photon emission computed tomography; SDS: summed difference score; SRS: summed rest score; SSS: summed stress score.

## Data Availability

Data available on request due to restrictions.

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
