# Peer review of "Automated Analysis vs. Expert Reading in Nuclear Cardiology: Correlations with the Angiographic Score"

_medicina, 2022, doi:10.3390/medicina58101432_

Round 1
Reviewer 1 Report
The SPECT MPI is a non-invasive imaging methodology for the investigation of patient’s CAD. The utilization of software packages in a Nuclear Medicine department for the analysis and the evaluation of the MPI outcomes is a common practice. However, the comparison of the final interpretations of clinicians with or without the utilization of SW scores and the correlation with the angiographic scores is a study of great interest concluding of the overall evaluation of the MPI examinations.
Moreover, some comments and points which should be under consideration are:
p.1, l.24 . It should be better if “acquired images” replaced by “Reconstructed images”. The same could be found in some further references into the text.
l.67. SPECT images should replaced by SPECT data.
p.1, l.29. It should added the acronym (ER) just after “expert reading”
Discussion should be renumbered as 5 & conclusion as 6.
Sec. 2.4.
It might be useful to be noted the injected activities (stress/rest) as well as their ranges to the patient’s group. Such data should be taken into account for the calculation/estimation of the effective doses in nuclear medicine and/or angiographic examinations.
The computation of the inter observer variability of the two nuclear medicine observers should be refereed. Did an agreement (numerical score) occur in all cases except the eight studies in which the third observer was utilized?
The “institutional adjustments” is a process and finally a feature that could influence greatly the categorization score of a study. The utilization of typical studies of normal and pathological cases analyzed by the specific clinical department, might be followed in order to improve the overall accuracy of the image-based characterization.
Sec.3.
Regarding the labelling of the characterization ability based on the Correlation coefficients ranges and their characterization [Turk J Emerg Med. 2018 Sep; 18(3): 91–93.]:
.00-.19 “very weak”, .20-.39 “weak” , .40-.59 “moderate” , .60-.79 “strong”, .80-1.0 “very strong”
In some clinical research articles “Moderate” positive correlation exists when the coefficient is up to 0.7 (0.5-0.7) [Malawi Med J. 2012 Sep; 24(3): 69–71.]
In the present study, the correlation coefficient characterized as moderate when its value is 0.3-0.5 and high (strong) when its value is more than 0.5. This is somehow highly optimistic. Thus, the readjustment the classes’ characterization should be accomplished.
Sec.4.
The 41.2 % in the angiographic score characterized as “0”- Normal. The computation of summed SSS, SRS and mainly the SDS in those studies might be consider. It is of great interest due to the suspicion that the MPI’s conclude in more False Positive interpretations.
The correlation between expert reading and SW packages (SRS & SDS) is limited (Table 4). The SDS values, that are the most critical indications, appear to be in the range of 0.56 to 0.68. and thus in the area of moderate. Thus, the clinical value of SW packages might be under consideration.
The correlation between angiographic score and experts reading is quite higher than most of the others (Table 5). This is a respectable outcome since it highlights the value of SPECT MPI examination. However, it should concern us the low score (less than .5) between angiographic and SW packages.
Sec.5
The readers make their interpretations based on the MPI images and on features extracted by the patients’ medical history (pre-test high risk patient, obesity, hypertension, breast artifact …).
The SW packages should be informed / updated for the critical data of each patient. That means that features such as the numerical BMI value… or the patterns that are utilized by the SW (pathological or normal) should be updated and corrected for the specific clinical environment. In the present study do we have a sense about the level of the above considerations?
Reviewer 2 Report
Dear Authors,
Please improve and extent your introduction
Please add more conclusion to your review
Kind regards
